# Pronounced Declines in Meperidine in the US: Is the End Imminent?

**DOI:** 10.3390/pharmacy10060154

**Published:** 2022-11-20

**Authors:** Lavinia R. Harrison, Rhudjerry E. Arnet, Anthony S. Ramos, Poul A. Chinga, Trinidy R. Anthony, John M. Boyle, Kenneth L. McCall, Stephanie D. Nichols, Brian J. Piper

**Affiliations:** 1Department of Medical Education, Geisinger Commonwealth School of Medicine, Scranton, PA 18509, USA; 2Department of Biology, Holy Family University, Philadelphia, PA 19114, USA; 3Department of Chemistry, University of Scranton, Scranton, PA 18510, USA; 4Department of Biology, University of Scranton, Scranton, PA 18510, USA; 5Department of Biology, Temple University, Philadelphia, PA 19112, USA; 6Department of Pharmacy Practice, Binghamton University, Johnson City, NY 13790, USA; 7Department of Pharmacy Practice, University of New England, Portland, ME 04103, USA; 8Center of Pharmacy Innovation and Outcomes, Forty Fort, PA 18704, USA

**Keywords:** opiate, pain, neurotoxicity, misuse

## Abstract

**Background**: Once a widely used analgesic in the United States (US), meperidine offered an alternative opioid to other opioids as a pain reliever and was widely assumed to be safer with acute pancreatitis. However, within the last two decades meperidine, has gone from a frequently used drug to being used only when patients exhibit atypical reactions to opioids (e.g., morphine and hydromorphone), to being taken off the World Health Organization List of Essential Medications and receiving strong recommendations for overall avoidance. The aim of this study was to identify changes in meperidine distribution in the US, and regional disparities as reported to the Drug Enforcement Administration’s Automation of Reports and Consolidated Orders System (DEA ARCOS) and Medicaid. **Methods**: Data related to meperidine distribution was obtained through ARCOS (2001–2021) and Medicaid public use files (2016–2021). Heat maps were used to visualize regional disparities in distribution by state. States outside a 95% confidence interval were statistically significant. **Results**: Meperidine distribution between 2001 and 2021 decreased by 97.4% (R = −0.97, *p* < 0.0001). There was a 34-fold state-level difference in meperidine distribution between Arkansas (16.8 mg/10 persons) and Connecticut (0.5 mg/10 persons) in 2020. Meperidine distribution in 2020 was elevated in Arkansas, Mississippi, and Alabama. In 2021, meperidine distribution was highest in Arkansas (16.7 mg/10 persons) and lowest in Connecticut (0.8 mg/10 persons). Total prescriptions of meperidine as reported by Medicaid decreased by 73.8% (R = −0.67, *p* = 0.045) between 2016 and 2021. **Conclusion**: We observed a decrease in the overall distribution of meperidine in the past two decades, with a similar recent decline in prescribing it to Medicaid enrollees. The shortage of some parenteral formulations is an important contributor to these declines, however, the most likely explanation for this global decline in use is related to an increased recognition of safety concerns related to important drug interactions and a neurotoxic metabolite. This data may reflect plans to phase out the use of this opioid, especially in the many situations where safer and more preferred opioids are available.

## 1. Introduction

Although opioids are a highly effective class of drugs to reduce acute pain, their short and long-term risks are considerable, including potentially fatal respiratory depression and opioid use disorder. Meperidine was originally synthesized in 1939 as an anticholinergic agent. It was soon after discovered to have analgesic properties as an opioid agonist [1]. In addition to opioid and anticholinergic properties, meperidine is serotonergic via inhibition of the serotonin re-uptake transporter, resulting in increased intrasynaptic serotonin [2]. The most important pathway for meperidine metabolism involves n-demethylation by cytochrome P450 2B6, 2C19, and 3A4 enzymes to normeperidine [3]. Normeperidine is renally excreted with a half-life of 14 to 48 h [1]. The half-life is prolonged in patients with renal dysfunction, and repeated dosing causes a significant increase in C_max_ and can lead to neurotoxicity. Normeperidine neurotoxicity has been reported to cause tachycardia, hypertension, muscle tremors, and grand mal seizures [1]. The US Food and Drug Administration approved the use of meperidine for its analgesic properties in 1942 [1]. Meperidine was the most widely used opioid analgesic in the US in 1987, prescribed by approximately 60% of physicians for acute pain and by 22% for chronic pain [1].

In 1984, the tragic death of an 18-year-old woman, Libby Zion (1965–1984), encouraged clinicians to take a closer look at the safety of meperidine. Her admission to the New York hospital led to an unfortunate death due to respiratory arrest secondary to bilateral bronchopneumonia, which was a result of serotonin syndrome caused from the deadly interaction between meperidine and her antidepressant, phenelzine [4,5]. Phenelzine is a monoamine oxide inhibitor. Monoamine oxide metabolizes serotonin; hence, inhibition increases serotonergic tone. When meperidine and phenelzine are combined, both serotonin metabolism and serotonin re-uptake are inhibited, resulting in excess synaptic serotonin and serotonin toxicity, also called serotonin syndrome. In severe cases, when not treated promptly, serotonin toxicity is fatal. In the case of Libby Zion, she was on long-term phenelzine therapy, and was administered a dose of meperidine for “agitation and shivering” in the emergency department. Shortly after, she experienced increased restlessness, agitation, confusion, and extreme hyperpyrexia (42 °C). She went into respiratory arrest and died only 3 h after the administration of meperidine. This drug interaction has been considered “key to her death” [5]. The persistent advocacy of her attorney father, Sydney Zion, led to a grand jury probe and report published in 1986. This report indicted the culture and practices of graduate medical education by increasing physician resident wellbeing and ability to engage in self-care. This sequence of events resulted in limiting excessive working hours of residents and interns, and can be credited with enhancing graduate medical education. In addition to the Libby Zion serotonin syndrome case [6], the meperidine analog in the Barry Kidston (1954–1978) dopamine neurotoxicity case with desmethylprodine promoted further reconsideration of the utility of this once ubiquitous agent [7]. 

In 1992, guidelines indicated that meperidine was only to be used in healthy patients who have had adverse side effects from other opioids such as morphine and hydromorphone [8] (Table 1). Furthermore, the medical dogma that meperidine was less likely to cause Sphincter of Oddi spasm compared with morphine, and thus the first line for acute pain associated with pancreatitis, was dispelled [9]. Thus, in light of the lack of benefits over morphine and significant risks of adverse effects and drug interactions with meperidine, its risks clearly outweigh the benefits [9]. Meperidine was removed from the World Health Organization’s Model List of Essential medicines in 2003 [10] (Table 1). The American Geriatrics Society Beers Criteria voiced their opinion on the drug in 2012, recommending that meperidine use be avoided. In 2014, meperidine was ranked on the US Fit for The Aged (FORTA) List Expert Consensus Validation 1, which is an easy-to-use clinical tool for healthcare providers to support pharmacotherapy decisions of older patients [11]. Meperidine was ranked on the FORTA scale in their most restricted category as a Class D drug, which is defined as a drug that should be avoided in older people, if possible and to use an alternative substance [11]. It was noted that meperidine has “high risk of seizures with repeated dosing. Poor efficacy with oral administration; increases delirium” [11]. In 2019 the American College of Obstetrics and Gynecology indicated that the use of meperidine generally is not recommended for peripartum analgesia [12] (Table 1). 

Meperidine is currently categorized in the US as a Schedule II-controlled substance with a “high potential for abuse”, with use potentially leading to severe psychological or physical dependence [13]. A prior pharmacoepidemiology report characterized changes in US meperidine distribution and use between 2001 and 2019. The total distribution of meperidine, as reported to the DEA, decreased precipitously between 2001 and 2019 [13]. However, four southern states (Arkansas, Alabama, Oklahoma and Mississippi) showed the largest meperidine distribution when correcting for the population [13]. Our goal here was to include updated information (2020 and 2021) as well as examine the changes among Medicaid recipients.

**Table 1 pharmacy-10-00154-t001:** Guidelines and other influential documents with cautions on the use of meperidine.

Year	Document	Information
1992	Agency for Health Care Policy and Research [8]	Meperidine should be used only for very brief courses in otherwise healthy patients who have demonstrated an unusual reaction like local histamine release at the infusion site or an allergic response during treatment with other opioids such as morphine or hydromorphone
2002	World Health Organization Essential Medications [14]	Pethidine (meperidine) listed in injection and tablet formulations
2003	World Health Organization Essentials Medications [10]	Pethidine (meperidine) no longer listed
2004 2012	Institute of Safe Medication Practices, Canada [15] American Geriatrics Society Beers Criteria [16]	Oral meperidine should be removed from the formulary of healthcare facilities. Parental meperidine should have a limited duration of 48 h. Avoid use of meperidine among elderly. Strong recommendation based on high-quality evidence to avoid meperidine. Not an effective oral analgesic in dosages commonly used, may cause neurotoxicity, safer alternatives available
2015	American Geriatrics Society Beers Criteria [17]	Strong recommendation based on moderate quality evidence to avoid meperidine, especially in individuals with chronic kidney disease. Not effective oral analgesic in dosages commonly used; may have higher risk of neurotoxicity, including delirium, than other opioids, safer alternatives available
2019	American Geriatrics Society Beers Criteria [18]	Strong recommendation based on moderate quality evidence to avoid meperidine. Not effective in dosages commonly used; may have higher risk of neurotoxicity, including delirium, than other opioids, safer alternatives available
2019	American College of Obstetrics and Gynecology [12]	The use of meperidine generally is not recommended for peripartum analgesic because its active metabolite, normeperidine, has a prolonged half-life in adults and a half-life of up to 72 h in the neonate

## 2. Materials and Methods

### 2.1. Procedures

Retail drug summary reports for all 50 states in 2020 and 2021 were obtained using the DEA’s Automated of Reports and Consolidated Orders System (ARCOS) [13,19]. Although ARCOS reports on controlled substances by weight (rather than more standard measures like prescriptions), this has shown high correspondence with state Prescription Drug Monitoring Programs [13,19]. Medicaid Prescriber Public Use Files were obtained for all 50 state totals in 2016–2021 [20]. 

### 2.2. Data Analysis

The total distribution of meperidine per state as reported to the DEA between 2001 and 2021 was corrected from population estimates from the American Community Survey for each year and state. States were ranked, and values outside of 1.96 standard deviations from the average were considered statistically significant. The 1.96 value was selected as only 5% of the area under a Gaussian curve are outside of this cutoff. Two choropleth maps were created for 2020 and 2021 using JMP statistical software and waterfall graphs were created using GraphPad Prism to visualize disparities in distribution. Linear regression was created using Prism to present the total distribution of meperidine by year. This was also completed for Medicaid prescriptions for 2016–2021.

## 3. Results

National meperidine distribution showed pronounced declines from 2001 to 2021 (Figure 1A). Similarly, prescriptions to Medicaid patients were converted to weight and showed reductions from 2016 until 20121 (Figure 1B).

Figure 2A shows an over thirty-fold difference in meperidine distribution in 2020 with three southern states, Arkansas, Mississippi, and Alabama, falling outside of a 95% confidence interval. Many New England and mid-Atlantic states were the lowest in the US (Figure 2B).

Figure 3A shows an over twenty-fold difference in meperidine distribution in 2021 with southern states (Arkansas, Mississippi, Alabama, and Louisiana) ranking in the top four. Many New England (Connecticut, New Hampshire, Maine, and Rhode Island) states were again the lowest in the US (Figure 2B).

## 4. Discussion

The United States had an overall decrease of 97.4% between 2001 and 2021 in meperidine. While it is no surprise that meperidine continued to decrease, the reported distribution of meperidine for 2021 was lower than expected. One key factor is declines in production, including the cessation of some formulations. Pfizer discontinued Demerol 100 mg/mL 20 mL vials in April 2021 [21]. In addition, Demerol 50 mg/mL 2 mL ampules are on long-term back order, and the company estimated a release date of February 2023 [22]. When comparing the use of meperidine in the US to those of Canada, there was an overall decrease in the use of oral meperidine from 2005 to 2010, but a ten-fold difference between the highest (Newfoundland) and lowest (Quebec) provinces [23].

Although meperidine distribution in the US continues to decrease, there was a 34-fold difference between the highest and lowest states in 2020. Codeine, another low-potency opioid, showed a four-fold difference, and oxymorphone the largest meperidine distribution [23,24]. Arkansas [16.8 mg/10 persons], Mississippi [15.3 mg/10 persons], and Alabama [12.7 mg/10 persons] were the highest distributors in 2020 and were significantly elevated relative to the national average. This is consistent with earlier data [13]. Similar to 2019, in 2020 meperidine varied by geographic region with south-central states, and those with more obesity, showing greater distribution [13]. As of 2020, Arkansas (36.4%, ranked number 9), Mississippi (39.7%, ranked number 1), and Alabama (39%, ranked number 3) was ranked as the top ten highest adult obesity rates in the US [25]. Colorado has the lowest at 24.2%, and was in the lower ranking distributors at number 43 [25] (Figure 2A).

The signature benefit of meperidine usage over other opioids was that it was purported to not cause spasms in the Sphincter of Oddi. Once this was disproven, the adverse effects of increased seizure risks and serotonin syndrome greatly outweighed any benefits of use for this weak opioid [10] (Table 1). Meperidine now primarily serves as an option for patients who may have special considerations, such as allergies to morphine or those that cannot use fentanyl or other alternatives. Although it may still be a bit premature, it is tempting to speculate whether meperidine will remain a rarely used Schedule II substance like cocaine [26] or methamphetamine [27], or join the ranks of heroin and propoxyphene which were discontinued in the US.

This study is not without limitations. Although in modest amounts, meperidine is also distributed by veterinarians to non-human animals, and this is reported to ARCOS. However, the use of meperidine at veterinary teaching institutions was negligible [28], and the temporal pattern in Medicaid was similar to ARCOS. Although controlled substances can be sent by mail order pharmacies across state lines beginning in 2020, there was a high degree of similarity in state rankings of meperidine distribution for 2019 [13] and 2020, so we believe this had little impact on our findings of pronounced geographic differences. Further, this pharmacoepidemiological report does not contain detailed patient-level information including the contribution of neuropsychiatric problems such as anxiety, depression, personality disorders, individual differences in cognitive function, psychoses, and sleep disorders, as well as medical co-morbidities and surgical histories, to meperidine distribution and prescribing. The contributions of bio-psycho-social factors, including past surgeries and neuropsychological histories to pain management, should be further explored in future investigations with electronic medical records, particularly in southern states with elevated meperidine use. Socio-legal and economic factors like continuing medical education for providers, prior authorizations, and co-pays will continue to be influential in creating a more judicious opioid prescribing culture.

## 5. Conclusions

Recent pharmacoepidemiologic studies of opioid use and regional distribution patterns are vital to informing opioid stewardship programs. This research can be accessed to develop intervention plans to phase out the use of meperidine in situations when it is not necessary. This can be done by producing a steady supply of alternatives and working with state agencies to make other options more readily available. Intervention plans may involve educating providers, including anesthesiologists, in continued use areas on the lack of benefits of prescribing meperidine. The use of meperidine will continue to decrease use as further understanding of its lack of superiority over other opioids and increased elevated risks reaches all prescribers in the US and beyond.

## Figures and Tables

**Figure 1 pharmacy-10-00154-f001:**
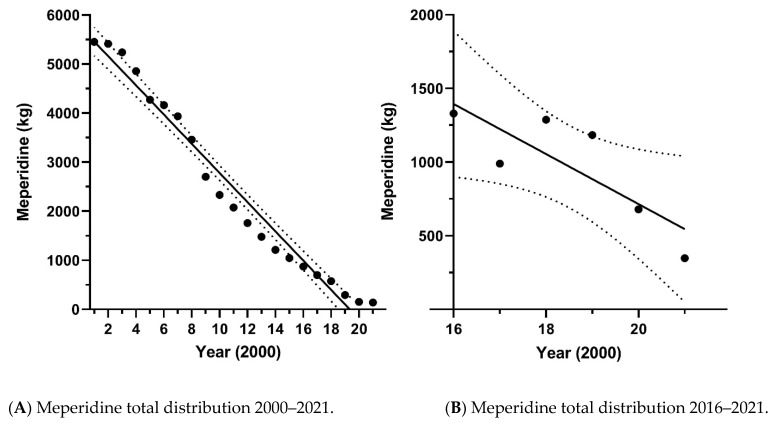
Total distribution of meperidine, as reported to the US Drug Enforcement Administration by the Automated Reports and Consolidated Orders System between 2001 and 2021, decreased by 97.4%. A linear regression over time was significant (R = −0.97, *p* < 0.0001) (**A**). The total distribution of meperidine, as reported by Medicaid Prescriber public use file between 2016–2021 decreased by 73.8%. A linear regression of national distribution over time was significant (R = −0.67 *p* = 0.045) (**B**).

**Figure 2 pharmacy-10-00154-f002:**
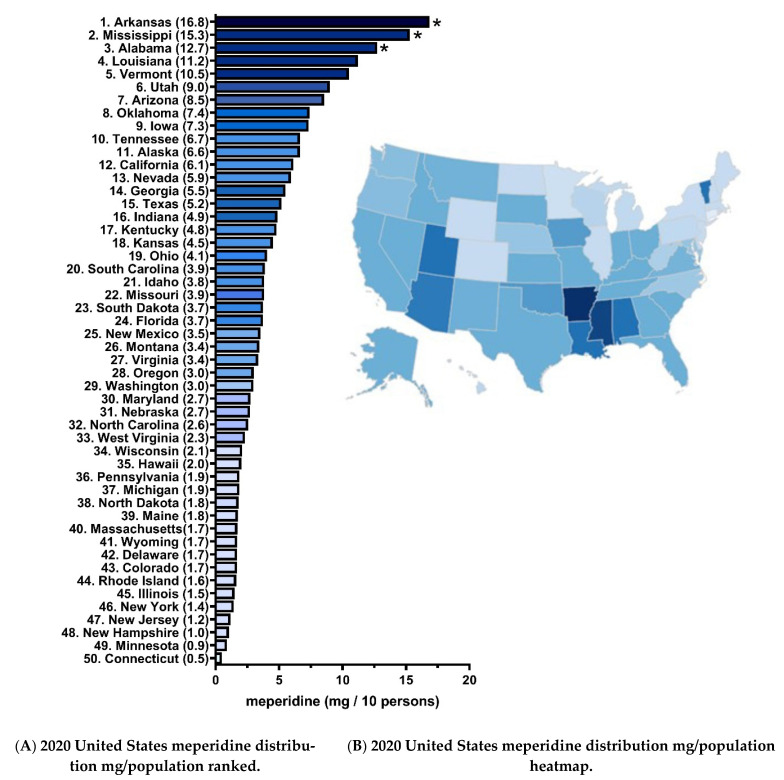
Waterfall plot (**A**) and heatmap (**B**) of meperidine distribution in 2020 as reported to the US Drug Enforcement Administration’s Automated Reports and Consolidated Orders System was highest in Arkansas (16.8 mg/10 persons) and lowest in Connecticut (0.5 mg/10 persons), a 34-fold difference. Arkansas, Mississippi, and Alabama had significantly elevated meperidine relative to the state average (mean 4.5 ± 3.6). *state outside ± 1.96 standard deviations.

**Figure 3 pharmacy-10-00154-f003:**
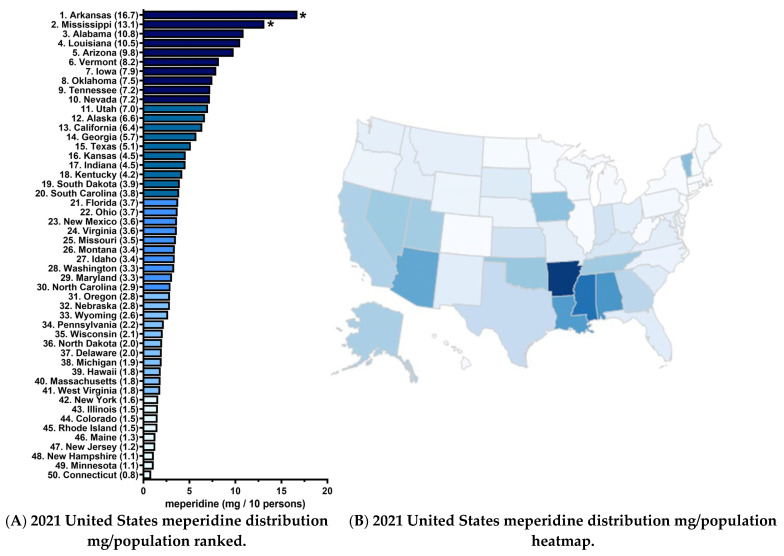
Waterfall plot (**A**) and heatmap (**B**) of meperidine distribution in 2021 as reported to the US Drug Enforcement Administration’s Automated Reports and Consolidated Orders System. Meperidine distribution was highest in Arkansas (16.7 mg/10 persons) and lowest in Connecticut (0.8 mg/10 persons), a 20.8-fold difference. Arkansas and Mississippi had significantly elevated meperidine relative to the state average (mean 4.4 ± SD 3.3). * state outside ± 1.96 standard deviations.

## Data Availability

All data is publicly available from the US Drug Enforcement Administration and the Centers for Medicare and Medicaid Services [13]. Extracted information is available at MedRxiv.

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
