# Peer review of "Pronounced Declines in Meperidine in the US: Is the End Imminent?"

_pharmacy, 2022, doi:10.3390/pharmacy10060154_

Round 1

Reviewer 1 Report

Once a widely used opioid in the United States, meperidine has increasingly fallen out of favor as an analgesic over the past two decades. The aim of this study was to identify changes in meperidine distribution in the United States and regional disparities as reported to the Drug Enforcement Administration’s Automation of Reports and Consolidated Orders System (DEA ARCOS) and Medicaid. Researchers observed a statistically significant decrease in the overall distribution of meperidine in the past two decades (2001-2021, 97.4%) with similarly appreciable declines in total prescriptions of meperidine to Medicaid enrollees (2016-2021, 73.8%). Additionally, researchers found regional disparities in meperidine distribution by state, with a notable 34-fold difference between the highest and lowest states (i.e., Arkansas and Connecticut) in 2020. In general, these findings are consistent with earlier data from a pharmacoepidemiology report that characterized changes in US meperidine distribution and use between 2001 and 2019.

The reviewer offers the following comments, by section of the manuscript, for the authors’ consideration.

Introduction

The sixth sentence of the opening paragraph (page 2, lines 49-51) regarding meperidine being categorized as a schedule II-controlled substance is superfluous and disrupts the flow of the historical nature of this paragraph. The authors should consider omitting this sentence.

Materials & Methods 

In the Data Analysis section (page 2, lines 82-82), the researchers indicate that states were ranked and values outside of 1.96 standard deviations from the average were considered statistically significant. Please provide additional context to explain how and/or why the researchers chose this cut-off value for state ranking, as it may not be obvious to readers.

Results

Although Figures 2 and 3 (page 4 and 5) present similar data but differ by year (2020 vs. 2021), the figure titles are slightly different. Specifically, Figure 3 includes the words “ranked” and “heatmap” in the title whereas Figure 2 does not. The authors should consider making the figure titles the same, for consistency, unless there is a compelling reason to the contrary.

Author Response

Attached and pasted below:

Introduction:  The sixth sentence of the opening paragraph (page 2, lines 49-51) regarding meperidine being categorized as a schedule II-controlled substance is superfluous and disrupts the flow of the historical nature of this paragraph. The authors should consider omitting this sentence.

Response: This content was removed from this paragraph to improve the historical flow. It was moved to a later paragraph.

Materials & Methods: In the Data Analysis section (page 2, lines 82-82), the researchers indicate that states were ranked and values outside of 1.96 standard deviations from the average were considered statistically significant. Please provide additional context to explain how and/or why the researchers chose this cut-off value for state ranking, as it may not be obvious to readers.

Response: Added to the methods “The 1.96 value was selected as only 5% of the area under a Gaussian curve are outside of this cutoff.”

Results: Although Figures 2 and 3 (page 4 and 5) present similar data but differ by year (2020 vs. 2021), the figure titles are slightly different. Specifically, Figure 3 includes the words “ranked” and “heatmap” in the title whereas Figure 2 does not. The authors should consider making the figure titles the same, for consistency, unless there is a compelling reason to the contrary.

Response: Adjusted the Figure 2 and 3 captions as suggested.

Figure 2A name: 2020 United States meperidine distribution mg/population ranked.

Figure 2B name: 2020 United States meperidine distribution mg/population heatmap.

Reviewer 2 Report

I have several problems with this study. it's likely a poor economic and intellectually resourced states are just using up old stock piles of narcotics,  without anyone addressing  the main reason most narcotics are used is for neuropsychiatric problems ie  anxiety, depression ,  personality issues  , low cognitive function , sleep disorders , and medical co morbidities  ! Narcotics are just drugs for general pain  , they don't differentiate emotional or physical pain! Plus emotional pain can be more agonizing than having a limb cut off! And phantom pain can be more painful , as can depression and there's nothing there on blood / medical tests are typically insufficient  ! The triggers are personality disorders and traits and  general categories of dozens of types of anxiety disorders and mixed depression states and even bipolar and psychotic patient perceptions of pain , life situations sleep,  medical etc . 

Narcotics are routinely prescribed for emotional pain even post operatively with almost a shut the patient up approach ,  with almost no attempts to get objective brain check ,  inflammatory markers such as interleukin 6 with CRP and immune profiles . For example high T-helper suppressor ratio is very predictable of osteoarthritis which patients are naive about daily pain and it's management . Plus many individuals have osteoarthritis and joint damage  without coordinating health imaging and still will underuse Tylenol to ibuprofen , and worse they don't know how to use anti-inflammatories ie protect their stomach, and don't use lifestyle techniques ie exercise to stretching / yoga awareness is low !  The natural steroids such as estrogen,  progesterone,  testosterone, DHEA are under utilized as aging Americans become more and more obese and in pain so they Don't realize exercise will reduce pain instead tend to get pain relief from drugs  including narcotics from the doctor! the complaint of pain has almost no evaluation Neuropsychiatric in medical evaluation including but not limited to thorough objective scales or instruments of pain or even subjective scales of pain locations frequency ( which changes unreliably with patience ) with almost no ipsative data to check patient reliability and validity of their information before treatment! Patients get pain drugs for essentially  boo-boos of pain ! And their post surgical course is difficult because of the mixture of surgical pain with pain perception disorders  from sleep problems to Psychiatric-disorders  , from neurological disorders ie concussion to painful medical conditions that remain undiagnosed ie diabetic neuropathy to cardiac angina  . Plus there is cognitive disabilities  and decreased  memory/ attentions  abilities that confuse the perception of pain . Patients with low cognitive functions can't keep up with their job and intellectual demands of aging and life basically like children say I have a boo-boo !

so to me it's no surprise that the poor Old Dixie patients continue to use outdated  or inferior narcotics ( Demerol 20 times more ) ,  but what most upsetting is the result of poor health is similar to the north with pandemics of obesity , neuropsychiatric decline , increases in dementia and crime ! and common non-professional citizens continue to have subpar care but the richer union of the north doesn't seem to display that highly inspired sense of health considering the higher use of so called better narcotics. So to me what's of interest is this - what is the neuropsychiatric cognitive state of these patients that are still using narcotics rather than antidepressants, or  exercise mood enhancer's , and hormone balancing   and comprehensive comorbid codependent variable treatment of their health needs ! At the minimum this paper needs to address some information about the psychiatric rates or surgical rates around the use of narcotics in those regions being a evaluated.

Some of the comments might leave less sophisticated readers to believe the journal is  endorsing the current prescribing methods around post surgical pain which are poorly controlled even when compared to the most generous approaches of the various state CME training boards that I take ( I have six state licenses and most of them have a different opioid training course ) and opioids are still hurting longitudinal health care and productivity in the United States.furthermore the differences in use of Demerol and other narcotic prescribing differences does not take into account economics of the pill's co pay and training of Physicans !

Author Response

Attached and pasted below. This includes Reviewer #2 and other changes that were made to expand beyond a short-report.

11 November 2022

Dear Editor Schommer,

Thank you for the opportunity to complete the minor revisions on our manuscript “Pronounced Declines in Meperidine in the US: Is the end imminent? “. These are described below. This includes greatly expanding the word count from under 1,500 to closer to 3,000. These modifications are further described below.

Please note that the US National Institutes of Health recommended that researchers begin posting preprints of their manuscript which we did at MedRxiv. This artificially inflates the similarity index when plagiarism detection software generates a false positive (frankly, we are surprised that the score wasn’t higher!). It would be helpful if the Managing Editor kept this complexity in mind in assessing the originality of this revision.

Together, we feel that this paper will be of substantial interest to the readership of Pharmacy.

Thank you,

Brian Piper

Reviewer #2:

  1. I have several problems with this study. it's likely a poor economic and intellectually resourced states are just using up old stock piles of narcotics, without anyone addressing  the main reason most narcotics are used is for neuropsychiatric problems ie  anxiety, depression ,  personality issues  , low cognitive function , sleep disorders , and medical co morbidities  ! Narcotics are just drugs for general pain  , they don't differentiate emotional or physical pain! Plus emotional pain can be more agonizing than having a limb cut off! And phantom pain can be more painful , as can depression and there's nothing there on blood / medical tests are typically insufficient  ! The triggers are personality disorders and traits and  general categories of dozens of types of anxiety disorders and mixed depression states and even bipolar and psychotic patient perceptions of pain , life situations sleep,  medical etc . 

Response: These are great insights! Added to line 222-224: “Further, this pharmacoepidemiological report does not contain detailed patient level information including the contribution of neuropsychiatric problems including anxiety, depression ,  personality disorders, individual differences in cognitive function , psychoses, and sleep disorders , as well as medical co-morbidities and surgical histories, to meperidine distribution and prescribing.

  1. Narcotics are routinely prescribed … At the minimum this paper needs to address some information about the psychiatric rates or surgical rates around the use of narcotics in those regions being a evaluated.

Response: Yes, that would be nice. Unfortunately, that might be beyond the scope of this project. Added to line 225-6: “The contributions of biopsychosocial factors including surgical, psychiatric, and neuropsychological histories to pain management should be further explored in future investigations with electronic medical records, particularly in southern states with elevated meperidine use.”

  1. Some of the comments might leave less sophisticated readers to believe the journal is  endorsing the current prescribing methods around post surgical pain which are poorly controlled even when compared to the most generous approaches of the various state CME training boards that I take ( I have six state licenses and most of them have a different opioid training course ) and opioids are still hurting longitudinal health care and productivity in the United States. furthermore the differences in use of Demerol and other narcotic prescribing differences does not take into account economics of the pill's co pay and training of Physicans !

Response: We added to line 229-231: “Socio-legal and economic factors like  continuing medical education for providers, prior authorizations, and co-pays will continue to be influential in creating a more judicious opioid prescribing culture.”

Other changes

Line 6 Title: Adjusted an affiliation:

3Department of Biology, Holy Family University, Philadelphia, PA                                                                      

Abstract Changes:

Lines: 22-23 changed added ‘mg’ to Arkansas (16.8 mg/10 persons) and Connecticut (0.5 mg/ 10 persons).

New Lines: Results: Meperidine distribution between 2001 and 2021 decreased by 97.4% (R=.-97, P < .0001). There was a 34-fold state-level difference in meperidine distribution between Arkansas (16.8 mg/10 persons) and Connecticut (0.5 mg/ 10 persons) in 2020. 

Introduction Changes:

Reviewer One recommended:  ‘The sixth sentence of the opening paragraph (page 2, lines 49-51) regarding meperidine being categorized as a schedule II-controlled substance is superfluous and disrupts the flow of the historical nature of this paragraph. The authors should consider omitting this sentence.’

Response: Lines 50-51 deleted as suggested.

Lines 51-55: Included more information to support the Libby Zion case and its connection to meperidine.

In 1984 the tragic death of an 18-year-old woman, Libby Zion, encouraged clinicians to take a closer look at meperidine. Her admission to the New York hospital led to an unfortunate death due to cardiac arrest from the deadly in-teraction between meperidine and her antidepressant, phenelzine [4]. The persistent advocacy of her father, Sydney Zion (1933 – 2009), regarding limiting excessive working hours of residents and interns can be credited with enhancing graduate medical education.

Original sentence Lines 55-57: Additionally, the medical dogma that meperidine was less likely to cause Sphincter of Oddi spasm compared with morphine, and thus the first line in acute pain associated with pancreatitis, was dispelled [6].’

Changed to Lines 58-59 ‘In addition to the Libby Zion serotonin syndrome case [5], the meperidine analog in the Barry Kidston (1954 - 1978) dopamine neurotoxicity case with desmethylprodine promoted further reconsideration of the utility of this once ubiquitous agent [6].’

Original Lines 61-62  ‘In 2019 the American College of Obstetrics and Gynecology indicated that the use of meperidine generally is not recommended for peripartum analgesia (Table 1).’ Moved to Lines 63-64

Lines 66-70 Included information about meperidine use in older age patients.

In 2014, meperidine was ranked on the US Fit for The Aged (FORTA) List Expert Consensus Validation 1, which is an easy-to-use clinical tool for health care providers to support pharmacotherapy decisions of older patients [8]. Meperidine was ranked on the FORTA scale in their most restricted category as a Class D drug, which is defined as a drug that should be avoided in older people if possible and to use an alternative substance [8]. It was noted that meperidine has “high risk of seizures with repeated dosing. Poor efficacy with oral administration; increases delirium” [8].

Lines 72-73: Added current meperidine guideline recommendation

‘Meperidine is currently categorized in the US as a schedule II-controlled substance with a “high potential for abuse”, with use potentially leading to severe psychological or physical dependence [4].’

Lines 83-84: Supplemental table moved to introduction and named Table 1.                                                                          

Material and Methods:

No changes

Results:

Linea 137-138: Added calculation of difference .

‘Meperidine distribution was highest in Arkansas (1.67 mg/10 persons) and lowest in Connecticut (0.08 mg/10 per-sons), a 20.8 fold difference.’

Supplemental Table: Lines 172-173 moved to lines 83-84 -Moved to introduction as Table 1. The citations were incorporated into the manuscript.

Discussion:

No other changes

Conclusions:

No other changes

Reference:

Reference Number 2 Added:

Murray JL, Mercer SL, Jackson KD. Impact of cytochrome P450 variation on meperidine N-demethylation to the neurotoxic metabolite normeperidine. Xenobiotica. 2020 Feb;50(2):209-222.

Reference Number 3 Added:

Latta KS, Ginsberg B, Barkin RL. Meperidine: a critical review. Am J Ther. 2002 Jan-Feb;9(1):53-68.

Reference

Reference Number 4 Added:

Waitman J, McCaffery M. Meperidine--a liability. Am J Nurs. 2001 Jan;101(1):57-8.

Reference Number 8 Added:

Pazan F, Gercke Y, Weiss C, Wehling M, Marcum ZA, Gokula M, et al. The US-FORTA (Fit fOR The Aged) list: Consensus validation of a clinical tool to improve drug therapy in older adults. J Am Med Dir Assoc. 2020; 21(3):439-e9.

Relocated references to 10-17

  1. Clinical Practice Guideline Number 1: Acute pain management: Operative or medical procedures and trauma. Clinical practice guideline. Rockville, Md: Agency for Health Care Policy and Research, US Dept of Health and Human Services; 1992. AHCPR publication 92-0032.

  1. WHO Model List (revised April 2002). Accessed 11/27/2020 at: https://apps.who.int/iris/bitstream/handle/10665/67335/a76618.pdf;jsessionid=EB8AA9ECBECA1F01743B09AC720AD91?sequence=1
  2. WHO Model List (revised April 2003). Accessed 11/27/2020 at: https://apps.who.int/iris/bitstream/handle/10665/68168/a80290.pdf?sequence=1

  1. ISMP Canada Safety Bulletin – Meperidine (Demerol®): Issues in medication safety; 2004. http://www.ismp-canada.org/download/safetyBulletins/ISMPCSB2004-08.pdf. Accessed September 13, 2022.

  1. American Geriatrics Society 2012 Beers Criteria Update Expert Panel. American Geriatrics Society updated Beers Criteria for potentially inappropriate medication use in older adults. J Am Geriatr Soc. 2012;60(4):616-631.

  1. American Geriatrics Society Beers Criteria Update Expert Panel. American Geriatrics Society 2015 updated Beers Criteria for potentially inappropriate medication use in older adults. J Am Geriatr Soc. 2015;63(11):2227-2246.

  1. 2019 American Geriatrics Society Beers Criteria Expert Panel. American Geriatrics Society 2019 Updated Beers Criteria for potentially inappropriate medication use in older adults. JAGS 2019; 67:674-694.

  1. American College of Obstetricians and Gynecologists (ACOG). ACOG practice bulletin no. 209: obstetric analgesia and anesthesia. Obstet Gynecol. 2019;133(3):e208-e225.
